# Linking Organic Metabolites as Produced by *Purpureocillium Lilacinum* 6029 Cultured on Karanja Deoiled Cake Medium for the Sustainable Management of Root-Knot Nematodes

**Abhishek Sharma [1,\*]**, **Aditi Gupta [2]**, **Manu Dalela [2]**, **Satyawati Sharma [2]**, **R. Z. Sayyed [3]**, **Hesham Ali El Enshasy [4,5,6] and Elsayed Ahmed Elsayed [7,8]**

[1]   Amity Food and Agriculture Foundation, Amity University Uttar Pradesh, Noida 201313, India
[2]   Centre for Rural Development and Technology, Indian Institute of Technology Delhi, Hauz Khas,
      New Delhi 110016, India; 29.aditi@gmail.com (A.G.); mannuiitd@gmail.com (M.D.);
      satyawatis@hotmail.com (S.S.)
[3]   Department of Microbiology, PSGVP Mandal's Arts, Science, and Commerce College,
      Maharashtra 425409, India; sayyedrz@gmail.com
[4]   Institute of Bioproduct Development (IBD), Universiti Teknologi Malaysia (UTM), Johor Bahru,
      Johor 81310, Malaysia; henshasy@ibd.utm.my
[5]   School of Chemical and Energy Engineering, Faculty of Engineering, Universiti Teknologi Malaysia (UTM),
      Johor Bahru, Johor 81310, Malaysia
[6]   City of Scientific Research and Technology Applications (SRTA), New Burg Al Arab, Alexandria 21934, Egypt
[7]   Zoology Department, College of Science, King Saud University, Riyadh 11451, Saudi Arabia;
      eaelsayed@ksu.edu.sa
[8]   Chemistry of Natural and Microbial Products Department, National Research Centre, Cairo 12622, Egypt
**\***   Correspondence: asharma5@amity.edu

**Abstract:**   Root-knot nematodes pose a serious threat to agriculture and forest systems, causing significant losses of the crop worldwide. A wide range of chemical nematicides has traditionally been used to manage phyto-nematodes. However, due to their ill effects on the environment, biological control agents (BCAs) like *Purpureocillium lilacinum* that exhibit antagonistic effects on root-knot nematodes are preferred. The current study focused on identifying nematicidal metabolites produced by the fungus *Purpureocillium lilacinum* cultivated on akaranja deoiled cake-based liquid medium through bioactivity-guided fractionation against *Meloidogyne incognita*. Column chromatography of the ethyl acetate extract of fungal filtrate exhibited the most potent fraction (fraction 14–15), giving 94.6% egg mass hatching inhibition on the 5th day and a maximum nematicidal activity of 62% against second-stage juveniles after 48 h at 5000 mg/L. Gas chromatography coupled with mass spectrometry (GC-MS) analysis of this fraction revealed five major compounds, viz., 2-ethyl butyric acid, phenyl ethyl alcohol, benzoic acid, benzene acetic acid, and 3,5-Di-t-butylphenol. These biocompounds have potential biocontrol applications in agriculture, but further in vivo studies are warranted.

**Keywords:**   bioassay; fourier transform infrared (FTIR); nematicidal; organic volatiles; scanning electron microscopy

## 1. Introduction

Root-knot nematodes pose a significant threat to agriculture and forest systems owing to their short life cycle and a wide range of host preferences [1,2]. Considering the drawbacks of chemical

management and control strategies, biological control of nematodes has lately been accepted as analternative solution [3,4]. This encompasses microbial agents, especially fungi, that are antagonistic to root-knot nematodes [5,6]. The most common mode of action of such fungi is to secrete nematocidal metabolites that affect nematode viability [7,8]. Some of the key metabolites produced by the basidiomycetes include thermolides A and B, omphalotins, ophiobolins, bursaphelocides A and B, and illinitone A [8], while phomalactone, aurovertins D and F, paeciloxazine, a pyridine carboxylic acid derivative, and leucinostatins are known to be produced by ascomycetes [9]. Only a few fungal secondary metabolites have been screened for their potential nematocidal activities to date. Hence, more studies are required to confirm the metabolites from fungal natural product libraries as nematocidal [10].

*Purpureocillium lilacinum* (formerly *Paecilomyces lilacinus*) is primarily a saprophyte, being able to compete for and use a wide range of common substrates in soil [11]. The sedentary stages of the root-knot nematode are most vulnerable to *P. lilacinum*. The fungus can colonize nematode reproductive structures, thus destroying females, cysts, and eggs of root-knot nematodes, particularly *Meloidogyne incognita* [12,13].

Varied non-traditional substrates are continuously being researched and evaluated for their potential to support the growth of such entomopathogenic fungi. Non-edible deoiled cakes, such as neem (*Azadirachta indica*), castor (*Ricinus communis*), karanja (*Pongamia pinnata*), jatropha (*Jatropha curcas*), and mahua (*Madhuca indica*) are a rich source of micro- and macronutrients and are therefore regarded as suitable substrates for the growth of *P. lilacinum* [14,15]. Our previous study successfully established the role of karanja deoiled cake as a novel substrate for enhancing the growth and nematocidal properties of *P. lilacinum* 6029 [16]. We also successfully related the role of non-volatile compounds, viz., amino acids [17] and leucinostatins [18], to the antagonistic activity of fungus biocontrol. In recent years, volatiles have been established as a potent factor for killing nematodes by nematophagous fungi [19]. Therefore, it would be an interesting proposition to identify volatile nematocidal metabolites of *P. lilacinum* responsible for its pathogenicity. Hence, this work focuses on robust and efficient in-vitro bioassays that involve isolation and characterization of novel nematocidal secondary volatile metabolites, thereby opening up an opportunity to develop bio-formulations comprising bioactive compounds of *P. lilacinum*.

## 2. Materials and Methods

### 2.1. Microorganism, Karanja Cake, and Culture Filtrate Preparation

*Purpureocillium lilacinum* 6029 was procured from the Indian Type Culture Collection, Division of Plant Pathology, Indian Agricultural Research Institute (IARI), New Delhi, India. The strain was maintained on potato dextrose agar (PDA) slants at 4 °C. Oilseeds of karanja were randomly collected from the wild trees growing in Pratapgarh District, Uttar Pradesh, India, (25.9026° N, 81.7787° E), thereby negating any possibility of the presence of pesticides in seeds or deoiled cake.

The carbon–nitrogen (C/N) ratio of the karanja deoiled cake was determined by estimating C and N using a CHNanalyzer (Perkin Elmer 2400 Series II, USA). The deoiled cake's carbon, hydrogen, and nitrogen contents were 42.26, 6.37, and 4.87%. The original pH of the deoiled cake was 5.5. The optimized C/N ratio of 35.88 of the broth was adjusted with sucrose as a carbon source and to pH5.9 by 1 N NaOH /1 N HCl [14]. Spores of *P. lilacinum* 6029 harvested from the PDA slant surface weremixed with sterile distilled water (10 mL) with 0.1% (*w/v*) Tween 80 to prepare the spore suspension. This spore suspension ($10^8$ spores/mL) was used to inoculate subsequent flasks containing karanja deoiled cake-based broth. The fungal culture was incubated at 28 °C for two weeks. After the incubation, the broth was filtered through Whatman No. 1 filter paper to obtain cell-free culture filtrate.

## 2.2. Nematode Cultures

The initial inoculum of *M. incognita* was obtained from the Division of Nematology, IARI, New Delhi, India. The pure population of the nematodes was maintained on brinjal (*Solanum melongena*) roots growing in earthen pots with autoclaved soil at Micromodel Complex, IIT, New Delhi, India. Egg masses of *M. incognita* were handpicked from the brinjal roots using sterilized forceps. At the same time, second-stage juveniles were collected after three days of incubation of the egg masses in sterile distilled water at room temperature.

## 2.3. Extraction of Culture Filtrate Using Different Solvents

The culture filtrate (5 L) was exhaustively extracted with hexane at a 1:1 ratio. The non-polar hexane extract was subsequently concentrated to a thick yellow liquid (0.72 g). The aqueous portion was further extracted using ethyl acetate as the solvent. The ethyl acetate extract was again concentrated using a rotary evaporator at 45 °C to give 1.21 g of colorless liquid. The remaining aqueous fraction was concentrated through lyophilization.

## 2.4. In Vitro Bioassays of Solvent Fractions

### 2.4.1. Effect on Juvenile Mortality

The above hexane, ethyl acetate, and aqueous extracts were suitably diluted with distilled water containing 0.5% Tween 80 to final concentrations of 312.5–10,000 mg/L each. A 1.0 mL sample of test concentration was placed in 24-well plates, and 0.5 mL of water containing approximately 150–200 freshly hatched juveniles of *M. incognita* were added to each well. In-vitro growth chamber experiments were monitored to assess the nematocidal potential of the solvent fractions. The numbers of immobilized juveniles were counted after regular intervals of 12, 24, and 48 h using a stereoscopic microscope. Immobilized juveniles were subsequently transferred to sterile distilled water for an hour to ascertain their mortality. If they failed to regain their mobility and appeared straight, they were considered dead.

### 2.4.2. Effect on Egg Mass Hatching Inhibition

The percentage of inhibition of egg mass hatching was studied using 24-well plates, each containing 2 mL of the above-designed tested concentrations of the three extracts. Four egg masses of uniform size and color were added to each well. Egg masses placed in sterile distilled water served as the control. Numbers of hatched juveniles were counted daily using a stereomicroscope until the hatching ceased. Two hundred μL samples of ethyl acetate, hexane, and sterile distilled water with 0.5% Tween 80 were considered as separate blank controls. In the case of the chemical control, a commonly used nematocide, carbofuran (FMC Corporation, Philadelphia, US), was used at concentration of 50 μg/mL [20].

## 2.5. Fourier Transform Infrared (FT-IR) Spectroscopy Analysis

An infrared spectrum of ethyl acetate extract was studied using an FT-IR spectrophotometer (Perkin-Elmer 1600, Waltham, MA, USA). The experiment was designed to study the functional groups on the surface of the extract between the spectral range of 4000 and 400 cm$^{-1}$ at a scan speed of 16 cm/s. Washed and dried samples were kept on the diamond probe of the attenuated total reflectance (ATR) for the spectroscopic analysis [21].

## 2.6. Identification of Nematocidal Volatile Compounds

The concentrated ethyl acetate fraction was subjected to bioassay-guided fractionation using a column packed with a silica gel (10 g; 60–80 mesh size; Merck, Germany) and chloroform; a methanol mixture (20:1; *v/v*) was the eluting solvent. The proportion of methanol was gradually increased, maintaining a gradient of 10:1, 4:1, and, finally, 100% pure methanol [22]. A total of 32 fractions,

20 mL each, were collected. They were subjected to thin-layer chromatography performed on silica plates (TLC Silica gel 60 F254, Merck, Germany) using chloroform–methanol (8:1; *v/v*) as the solvent. Spots were developed by spraying 10% sulfuric acid and charring the plate at high temperatures (80–100 °C) in a hot air oven. Fractions giving similar spots were pooled into five groups and subsequently analyzed for their nematocidal potential using the in-vitro assays mentioned above.

The pooled fraction, Fr 14–15 (1.98 mg), showed maximum nematocidal activity and was subjected to gas chromatography–mass spectrometry (GC-MS). The GC-MS spectrum was recorded on a Shimadzu GC-MS-QP 2010 Plus mass spectrometer equipped with an Rtx-1MS column (30.0 m length, 0.32 mm i.d., and 0.25 μm film thickness). The column max temperature was 330 °C. The temperature of the injector port and detector were maintained at 260 °C and 270 °C, respectively. The oven temperature was maintained at 60 °C for 2 min and then increased to 230 °C at 2 °C/min for 12 min. Nitrogen was used as the carrier gas at a flow rate of 0.9 mL/min. The injected sample volume was 1 μL. The peaks were identified by comparing the synthetic standards mentioned in the NIST62-MS library (Shimadzu Corporation, Kyoto, Japan).

*2.7. Statistical Analysis*

All the experiments were performed in triplicate, and the data collected were expressed as mean value and standard deviation (SD). The results were subjected to a one-way analysis of variance (ANOVA) using SPSS for Windows (version 18.0). The significance of the difference was determined, according to Duncan's multiple range test (DMRT). *p* values < 0.05 were considered to be statistically significant. Effective lethal doses ($LC_{50}$ and $LC_{90}$) and chi-squared values were determined through the probit analysis method using StatPlus 2009 software [23].

## 3. Results

*3.1. In Vitro Bioassays of Solvent Fractions of P. lilacinum Filtrate*

At a tested concentration of 5000 mg/L, ethyl acetate extract showed maximal nematocidal activity. It bestowed 100% mortality within 48 h of the experimental period. With aqueous extract, the highest juvenile mortality of 89.2% was recorded with a 10,000 mg/L concentration at 48 h. On the other hand, hexane extract showed relatively poor nematocidal properties, with a maximum of only 29.7% at a 10,000 mg/L concentration after 48 h. Similarly, the ethyl acetate extract imparted (*p* < 0.05) nematocidal efficacy by completely inhibiting egg mass hatching with a tested concentration of 5000 mg/L on the 5th day of the experiment. The effect of aqueous extract was less effective, with 91.3% inhibition recorded for a 10,000 mg/L concentration, followed by 80.5% at a 5000 mg/L concentration. The hexane extract was again found to be the least effective, with hatching inhibition ranging from 0 to a mere 38.2% with a tested concentration range of 312.5 to 10,000 mg/L. In the chemical control, 100% mortality and egg mass inhibition were observed. In a distilled water +5% Tween 80 blank control, all the juveniles were motile at 48 h and exhibited complete egg mass hatching compared to 12.1 ± 2.3% and 13.2 ± 2.2% mortality and 2.1 ± 0.0% and 12.3 ± 0.2% inhibition obtained with the ethyl acetate and hexane blank, respectively (Table 1).

Regarding the toxicity of fungal filtrate, ethyl acetate extract exhibited intense nematocidal activity against second-stage juveniles and egg masses of root-knot nematodes with $LC_{50}$ values of 781.48 mg/L and 1004.58 mg/L, respectively (Table 2). Hexane extract showed fragile nematocidal activity, and hence its toxicity was not calculated. In contrast, the aqueous extract of *P. lilacinum* filtrate had $LC_{50}$ values of 2058.12 and 1556.24 mg/L against second-stage juveniles. The $LC_{50}$ values for nematocidal activity and egg mass hatching against second-stage juveniles of root-knot nematodes were 2490.56 mg/L and 3198.24 mg/L, respectively, compared to 9575.24 mg/L and 3198.06 mg/L obtained with aqueous extract of *P. lilacinum* filtrate, respectively (Table 2).

**Table 1.** Effect of different solvent extracts of *P. lilacinum* 6029 filtrates on mortality (mean ± SD) and egg mass hatching inhibition * (mean ± SD) of *M. incognita*.

| Concentration (mg/L) | Ethyl Acetate Extract | | Hexane Extract | | Aqueous Extract | |
|---|---|---|---|---|---|---|
| | Mortality (%) at 48 h | Egg Mass Hatching Inhibition (%) | Mortality (%) at 48 h | Egg Mass Hatching Inhibition (%) | Mortality (%) at 48 h | Egg Mass Hatching Inhibition (%) |
| 312.5 | 16.2 ± 2.6 $^b$ | 15.9 ± 1.2 $^b$ | 0.0 ± 0.0 $^a$ | 0.0 ± 0.0 $^a$ | 7.3 ± 0.0 $^a$ | 11.5 ± 0.0 $^a$ |
| 625 | 43.1 ± 1.8 $^c$ | 26.5 ± 1.0 $^c$ | 0.0 ± 0.0 $^a$ | 17.2 ± 1.2 $^b$ | 12.6 ± 1.1 $^b$ | 23.2 ± 1.9 $^b$ |
| 1250 | 76.9 ± 1.8 $^d$ | 51.7 ± 1.7 $^d$ | 10.8 ±1.8 $^b$ | 20.2 ± 1.8 $^c$ | 32.5 ± 1.9 $^c$ | 41.4 ±1.2 $^c$ |
| 2500 | 91.9 ± 2.8 $^e$ | 82.3 ± 2.2 $^e$ | 17.3 ± 1.6 $^c$ | 27.5 ± 1.9 $^d$ | 61.2 ± 2.6 $^d$ | 68.7 ± 1.8 $^d$ |
| 5000 | 100.0 ± 0.0 $^f$ | 100.0 ± 0.0 $^f$ | 23.9 ±1.2 $^d$ | 31.1 ± 1.6 $^e$ | 77.3 ± 1.2 $^e$ | 80.5 ± 1.3 $^e$ |
| 10,000 | 100.0 ± 0.0 $^f$ | 100.0 ± 0.0 $^f$ | 29.7 ± 1.9 $^e$ | 38.2 ± 2.8 $^f$ | 89.2 ± 2.1 $^f$ | 91.3 ± 0.9 $^f$ |
| DW+Tween blank | 0.0 ± 0.0 $^a$ | 0.0 ± 0.0 $^a$ | 0.0 ± 0.0 $^a$ | 0.0 ± 0.0 $^a$ | 0.0 ± 0.0 $^a$ | 0.0 ± 0.0 $^a$ |
| Ethyl acetate blank | 12.1 ± 2.3 $^b$ | 11.7 ± 1.1 $^b$ | Nd | Nd | 9.9 ± 0.1 $^a$ | 13.4 ± 0.2 $^a$ |
| Hexane blank | Nd | Nd | 2.1 ± 0.0 $^a$ | 12.3 ± 0.2 $^a$ | 11.2 ± 0.2 $^a$ | 15.3 ± 0.1 $^a$ |
| Chemical control ** | 100.0 ± 0.0 $^f$ | 100.0 ± 0.0 $^f$ | 100.0 ± 0.0 $^f$ | 100.0 ± 0.0 $^g$ | 100.0 ± 0.0 $^g$ | 100.0 ± 0.0 $^g$ |

Nd = Not defined. DW = Distilled water. The letters a, b, c, d, e, f, and g indicate significant differences between the results ofdifferent concentrations of each extract ($p < 0.05$) using DMRT. * Data recorded on 5th day. ** Carbofuran was used at a concentration of 50 μg/mL.

**Table 2.** Toxic effect of solvent extract of *P. lilacinum* filtrate against second-stage juveniles and egg mass hatching of *M. incognita*.

| Toxicity Factor | Ethyl Acetate Extract | | Aqueous Extract | |
|---|---|---|---|---|
| | Second Stage Juveniles | Egg Mass Hatching | Second Stage Juveniles | Egg Mass Hatching |
| $LC_{50}$ (mg/L) | 781.48 | 1004.58 | 2058.12 | 1556.24 |
| 95% Confidence intervals | (556.72–1040.83) | (725.94–1360.21) | (1785.83–2377.57) | (1333.46–1811.07) |
| $LC_{90}$ (mg/L) | 2490.56 | 3198.06 | 9575.24 | 8385.22 |
| 95% Confidence intervals | (1748.16–4599.24) | (2192.86–6184.47) | (7571.27–12877.9) | (6545.68–11495.39) |
| Chi-squared ($\chi^2$) | 10.929 | 12.136 | 2.528 | 1.318 |

* Concentrations ranged from 312.5–10,000 mg/L for both ethyl acetate and aqueous extract of *P. lilacinum* 6029 filtrate.

### 3.2. Identification of Nematocidal Volatiles

Since the ethyl acetate extract was found to be the most effective against *M. incognita*, it was subjected to FT-IR spectroscopy (Figure 1). The FT-IR spectrum showed the stretching vibration of C=O at 1713 cm$^{-1}$. An O-H stretching vibration was identified in the region of 2920 and 3026 cm$^{-1}$, which corresponds to the carboxylic group's presence. A broad peak at 1220 cm$^{-1}$ and a sharp peak at 920 cm$^{-1}$ were also observed, which correspond to either the benzene ring or the carbon–oxygen bond of the acid group present in the compound. A small but sharp peak at 1303 cm$^{-1}$ was observed, which corresponds to the stretching vibration of -OH groups present in the extract.

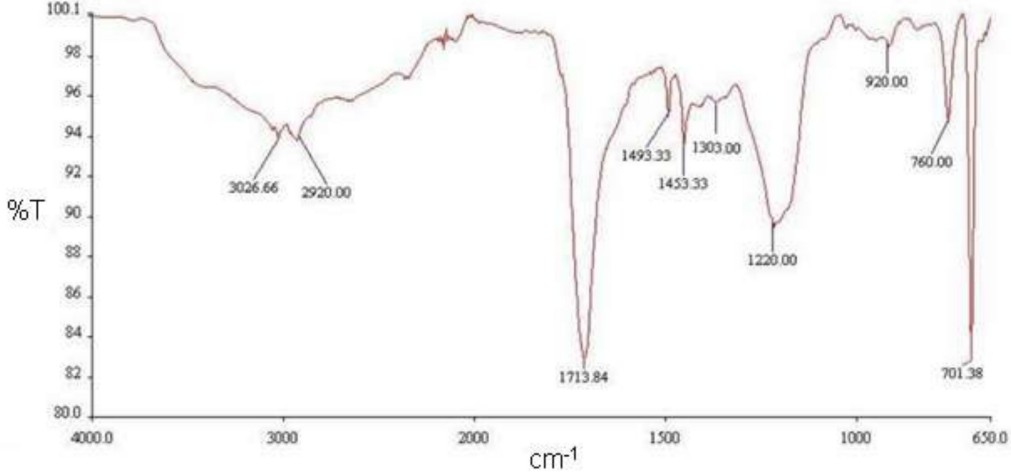

**Figure 1.** The FT-IR spectrum of ethyl acetate extract of *P. lilacinum* 6029 culture filtrate. Ethyl acetate extract of *P. lilacinum* 6029 culture filtrate was scanned in FT-IR in the range of 4000 and 400 cm$^{-1}$ at a scan speed of 16 cm/s and spectral analysis was performed with the help of a diamond probe of ATR.

The FT-IR study shows the presence of bioactive compounds with phenolic moiety in ethyl acetate extract. However, for precise identification of nematocidal compounds, the extract was further subjected to column chromatography followed by GC-MS.

Silica gel column chromatography of the extract yielded 32 fractions, which were later grouped into five fractions after examining thin-layer chromatography (Figure S1).

These fractions were further assayed at a concentration of 5000 mg/L for their bioefficacy using the methods described above. The pooled fraction, Fr 14–15, showed 94.6% egg mass hatching inhibition on the 5th day and a maximum nematocidal activity of 62% against $J_2$ after 48 h of the experiment (Figure 2). The data inspection disclosed the trend of results against $J_2$ as Fr 14–15 > Fr 16–18 > Fr 19–26 > Fr 28–32.

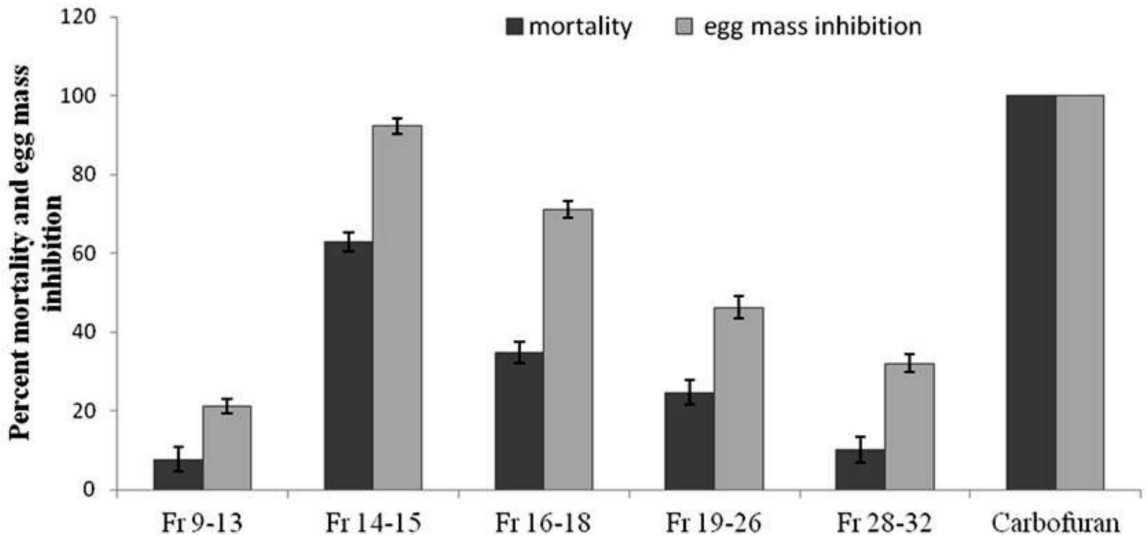

**Figure 2.** In-vitro egg mass hatching inhibition (after five days) and second-stage juvenile mortality (after 48 h) of *M. incognita* in response to ethyl acetate fractions of *P. lilacinum* 6029 filtrates. Vertical bars over the histogram indicate the standard deviation (SD). For a particular series, bars followed by different letters are statistically different from each other ($p < 0.05$) according to Duncan's multiple range test (DMRT). Ethyl acetate fractions (5000 mg/L) and carbofuran (50 µg/mL) and a blank were used to study the mortality of the juveniles and egg mass hatching inhibition.

The efficacy of egg mass hatching inhibition was in the order: Fr 14–15 > Fr 16–18 > Fr 19–26 = Fr 28–32. The pooled fraction, fraction 14–15, which displayed maximum efficacy, was further subjected to GC-MS analysis (Figures S2 and S3).

It contained more than 20 compounds, with five major ones shown in Table 3. The GC-MS analysis of Fr 14–15 revealedthe presence of a phenyl moiety to be a common feature in at least four of the primary compounds identified. The '3,5-Di-t-butyl-phenol was present in the highest amounts.

The pooled fraction, Fr 14–15 (1.98 mg), showed maximum nematocidal activity and was subjected to gas chromatography–mass spectrometry (GC-MS). The GC-MS spectrum was recorded on a Shimadzu GC-MS-QP 2010 Plus mass spectrometer equipped with an Rtx-1 MS column (30.0 m length, 0.32 mm i.d., and 0.25 µm film thickness). The column max temperature was 330 °C. The temperature of the injector port and detector were maintained at 260 °C and 270 °C, respectively. The oven temperature was maintained at 60 °C for 2 min and then increased to 230 °C at 2 °C/min for 12 min. Nitrogen was used as the carrier gas at a flow rate of 0.9 mL/min. The injected sample volume was 1 µL. The peaks were identified by comparing the synthetic standards mentioned in the NIST 62-MS library (Shimadzu Corporation, Japan).

**Table 3.** Chemical composition of ethyl acetate fraction (Fr 14–15) of *P. lilacinum* 6029 filtrates analyzed by GC-MS.

| Major Compounds | Retention Time (min) | Area (%) |
|---|---|---|
| 2-ethyl butyric acid | 8.21 | 0.75 |
| Phenyl ethyl alcohol | 8.75 | 1.55 |
| Benzoic Acid | 9.81 | 1.69 |
| Benzene acetic acid | 12.06 | 6.05 |
| 3,5-Di-t-butyl-phenol | 19.46 | 7.66 |

A 1 μL sample of ethyl acetate fraction (Fr 14–15) of *P. lilacinum* 6029 was injected in the Rtx-1 MS column, which was flowed with nitrogen gas at a flow rate of 0.9 mL/min. The resulting peaks were identified from NIST62-MS standard library.

## 4. Discussion

The bioassay of different solvent extracts of *P. lilacinum* 6029 filtrate showedvarious nematode antagonistic metabolites of a fungus, with the ethyl acetate fraction havingthe most promising effects against *M. incognita*. This suggests that volatiles withlethal actions are intermediary in polarity. The results of the present study corroborate the findings of Siddiqui et al. [24], in which ethyl acetate extract of *P. lilacinum* was found to be more lethal to *M. javanica* juveniles as compared to hexane extract. The low activity of each fraction assemblage compared to whole ethyl acetate extract suggested the secondary metabolites' synergistic role. Our results are in accordance with those obtained by Strobel et al. [25]. The mixture of volatiles of *Phomopsis* sp., when broken down into several classes of compounds, did not show the same antifungal effects.

The current analytical study of the ethyl acetate extract of the fungal filtrate by FT-IR spectroscopy and GC-MS demonstrated the presence of phenolic compounds possessing nematocidal activity. Gapasin et al. [26] analyzed the GC-MS spectrum of the active fraction of *P. lilacinum* filtrate and partially identified the main compound as a derivative of azulene containing two unsaturation points. We envisage that the utilization of karanja deoiled cake as a novel substrate could have triggered some metabolic pathways leading to the synthesis of phenolic metabolites, which were not known to be associated with *P. lilacinum*. Phenol is already known for its toxic effects on cells and has been usedasan antiseptic in clinical applications for a long time [27]. These findings suggest the possible role of the phenyl moiety in potent nematocidal compounds. Niu et al. [28] reported the "Trojan Horse" mechanism, which states that the volatile compounds secreted by the biocontrol agent attract the nematodes towards them. Various other virulent factors, like extracellular protease, kill the nematode by destroying the outermost protective barrier (cuticle) and modulating the nematode's immune system. It is worth mentioning here that the application of karanja deoiled cake as the substrate for *P. lilacinum* 6029 offers a suitable alternative for the disposal of natural organic materials, which otherwise could become environmental pollution.

The drawbacks of chemical measures for efficient nematode control must be overcome by intensifying the search for environmentally beneficial and cost-efficient alternatives. The current study contributes to the unexploited yet promising novel nematicides from the nematophagous fungus *P. lilacinum*, which could be used for the biocontrol of phytoparasitic nematodes. The metabolites isolated and identified by us (2-ethyl butyric acid, phenyl ethyl alcohol, benzoic acid, benzene acetic acid, and 3,5-Di-t-butylphenol) showed promising nematocidal potential. However, a detailed investigation of their potency under field conditions is warranted.

**Supplementary Materials:** The following are available online at http://www.mdpi.com/2071-1050/12/19/8276/s1, Figure S1: Thin-layer chromatography of 32 fractions of ethyl acetate extracts of *P. lilacinum* 6029 pooled into five fractions. The solvent system used for development is CHCl3: MeOH = 8:1, Figure S2: GC spectra of ethyl acetate fraction, Fr 14–15, of *P. lilacinum* culture filtrate, Figure S3: Mass spectra of five major compounds identified in the fraction, Fr 14–15.

**Author Contributions:** Conceptualization, S.S.; investigation and writing of the original draft of the manuscript, A.S.; methodology, A.G. and M.D.; reviewing and editing of the manuscript, R.Z.S. and H.A.E.E.; fund acquisition, E.A.E. All authors have read and agreed to the published version of the manuscript.

**Funding:** This research was funded in by Allcosmos Industries Sdn. Bhd. through research project No. R.J130000.7344.4B200 and UTM-RMC. The authors are also thankful for King Saud University, Riyadh, Saudi Arabia, for funding the work through the Researcher Supporting Project, Project No. RSP-2020/52.

**Acknowledgments:** The authors gratefully acknowledge the financial support provided by the National Oilseeds and Vegetable Oils Development (NOVOD) Board, Ministry of Agriculture and Farmers' Welfare, India), Gurgaon, India, Allcosmos Industries Sdn. Bhd., Malaysia, and Researcher Supporting Project (No. RSP-2020/52), King Saud University, Riyadh, Saudi Arabia.

**Conflicts of Interest:** The authors declare no conflict of interest.

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
