# Peer review of "Linking Organic Metabolites as Produced by Purpureocillium Lilacinum 6029 Cultured on Karanja Deoiled Cake Medium for the Sustainable Management of Root-Knot Nematodes"

_sustainability, doi:10.3390/su12198276_

Round 1

Reviewer 1 Report

Linking volatile organic metabolites as produced by Purpureocillium lilacinum 6029 for the sustainable management of root-knot nematodes

Dear Authors,

I think, overall, that the experiments were designed, mainly, properly and manuscript is well prepared but I have few questions and suggestions:

22: cuasing -> causing

24: chemical based nematocidal preparations -> chemical nematicides

27 (and in whole text): space between

31, 115, 215, graph 3: egg mass hatching -> we rather say that larvae  hatches, not eggs. Eggs are laid before

32: change in whole text e.g. 48h, 5000 mg/l (proper abbreviation record)

32 (and in whole text): correct all punctuation mistakes

45: it would be nice to see few examples

55: “micro” change into “micro-“

58: We also successfully related the role of non-volatile compounds viz., amino acids [15], and leucinostatins[16] towards the antagonistic activity of the biocontrol fungus ->  We also successfully related the role of non-volatile5 compounds viz., amino acids [15], and leucinostatins [16] towards the antagonistic activity of the  fungus biocontrol.

61: I do not think this is appropriate word here

73: estimating C, H and N (you forgot about H)

73: level?

78: you did not mention suspension before

78: Tween 80 not Tween-80

94, 95, 97: italics

106: It would be more reader friendly if this paragraph was divided into two sections, first: nematocidal effect on larvae and second: larvae hatching bioassay

107: it occurs that ethyl acetate extract was the most effective, it would be good to see what is the effect of ethyl acetate alone and then compare it with extract

113: add “water”

156: pictures should be below the text, its hard to compare text with picture in other option

160: please change the scale, now it is impossible to compare the size of spores

161: I would say, when looking at your pictures, that the shape is more regular but not oval. Please find better photo or change it

165: test->tested

201: b and d - the same picture - change it, on d - which spore is budding?

202: photo is indistinct, change into one with better quality

210: where is the negative control on this graph?

228: change into: Aerial spores whereas more constant in size, are smaller than blastospores.

237: you did not check action of individual fractions but complex of fractions (group composed of few fractions e.g. 9-13, 16-18 etc.) Fig. 3 so you cannot write it like that

242: you did not tested that! If you want to write something like that, first check it!

254: has been used as antiseptic

Author Response

I think, overall, that the experiments were designed, mainly, properly and manuscript is well prepared but I have few questions and suggestions:

22: cuasing -> causing

Response: Needful has been done. Thanks.

24: chemical based nematocidal preparations -> chemical nematicides

Response: Needful has been done. Thanks.

27 (and in whole text): space between

Response: Needful has been done. Thanks.

31, 115, 215, graph 3: egg mass hatching -> we rather say that larvae hatches, not eggs. Eggs are laid before

Response: Sir, to the best of our understanding, insects & nematodes lay eggs followed by their hatching into larvae. Larvae (or juveniles in nematodes) do not hatch but they are the outcome of egg hatch. Larvae molts in to pupae and adults (metamorphosis). I am attaching a link of the published article in the support of our claim(https://www.sciencedirect.com/science/article/pii/S0304380006005515).

32: change in whole text e.g. 48 h, 5000 mg/l (proper abbreviation record)

Response: Thanks for the minute detailing. The needful has been done.

32 (and in whole text): correct all punctuation mistakes

Response: Needful has been done. Thanks.

45: it would be nice to see few examples

Response: Thanks for the kind suggestion. Examples of metabolites produced by different group of fungi are now been added. We believe these examples would bring more clarity to the audiences. Thanks again.

55: “micro” change into “micro-“

Response: Needful has been done. Thanks.

58: We also successfully related the role of non-volatile compounds viz., amino acids [15], and leucinostatins[16] towards the antagonistic activity of the biocontrol fungus ->  We also successfully related the role of non-volatile5 compounds viz., amino acids [15], and leucinostatins [16] towards the antagonistic activity of the  fungus biocontrol.

Response: The statement has been rephrased as suggested. Thanks.

61: I do not think this is appropriate word here.

Response: The word “exciting” has now been replaced with “interesting”.Thanks.

73: estimating C, H and N (you forgot about H)

Response: The statement was in context to determining C/N ratio of the cake which can be calculated by estimating C and N content only. That is why we did not use H there. But yes, H is important and we did mention in the next statement. We have also now modified the name of the instrument from elemental analyzer to CHN analyzer to emphasize C, H and N estimation in our study.

73: level?

Response: CHN level of the cake are 42.26, 6.37, and 4.87%, respectively.

78: you did not mention suspension before

Response: The desired information has now been incorporated.

78: Tween 80 not Tween-80

Response: OK.

94, 95, 97: italics

Response:Complete manuscript have been rec-checked carefully and these and all others mistakes have been rectified. Thanks.

106: It would be more reader friendly if this paragraph was divided into two sections, first: nematocidal effect on larvae and second: larvae hatching bioassay

Response: Thanks for the suggestion. The paragraph has now been divided into two sub-sections.

107: it occurs that ethyl acetate extract was the most effective, it would be good to see what is the effect of ethyl acetate alone and then compare it with extract

Response: We appreciate the reviewer’s concern here. During the investigation, we had made sure that the solvent gets completely evaporated (using rotavapour) and then required concentrations were prepared by using distilled water and 0.5% Tween 80. Distilled water and 0.5% Tween 80 alone had taken as blank control and the same has been mentioned in the methodology section too.

113: add “water”

Response: Added.

156: pictures should be below the text, its hard to compare text with picture in other option

160: please change the scale, now it is impossible to compare the size of spores

161: I would say, when looking at your pictures, that the shape is more regular but not oval. Please find better photo or change it

Responses to 156, 160 and 161: As per the kind advice of another reviewer, the entire morphological study has now been removed from the article as this study was not directly related with the theme of the article.

165: test->tested

Response: The word ‘test’ has now been replaced with ‘tested’ in the result section.

201: b and d - the same picture - change it, on d - which spore is budding?

Response: The pictures have now been removed.

202: photo is indistinct, change into one with better quality

Response: The pictures have now been removed.

210: where is the negative control on this graph?

Response:In a blank control, there were nil mortality of the juveniles and egg mass hatching inhibition and hence did not show in the graph. The same has now been mentioned in the caption too. Thanks for the suggestion.

228: change into: Aerial spores whereas more constant in size, are smaller than blastospores.

Response: The entire discussion has been removed now in the revised manuscript.

237: you did not check action of individual fractions but complex of fractions (group composed of few fractions e.g. 9-13, 16-18 etc.) Fig. 3 so you cannot write it like that

Response: The statement has been rephrased.

242: you did not tested that! If you want to write something like that, first check it!

Response: The statements relating nematocidal property with the Karanja deoiled cake has now been removed.

254: has been used as antiseptic

Response: The needful has been done.

Reviewer 2 Report

In presented manuscript entitled "Linking volatile organic metabolites as produced by

 Purpureocillium lilacinum 6029 for the sustainable management of root-knot nematodes " authors analyzed nematocidal effect of various extracts obtained from P. lilacinum. Authors used non-typical substrate for cultivation Karanja deoiled cake, which should improve nematocidal activity as they reported previously. As this had probably high impact or achieved results I recommend adding this info to the title. On the other hand I am not sure that metabolites in extract were volatile, may be some of them, but not all so I suppose to remove this word from title.

In introduction as well as in discussion authors deal a lot with various substrates and their effect on fungal metabolites. This is very interesting but this is not analyzed in article. Comparison of metabolites from traditional cultivation and karanja based method should be performed. Otherwise author must to focus on rather on metabolites themself than on cultivation substrate. Comparison of substrates is a weakness of manuscript. Whole work is more exploratotory than hypothesis driven, but it looks to be actual.

Introduction is short and do not provide insight to metabolome of entomopathogenic/nematopathogenic fungi.

Why authors incorporate morphological study and SEM? I consider it to be redundant. Shape of spores is not important when we are talking about metabolites. According my opinion it should be removed from article (from methods, results and discussion)

In material and methods it is stated than solvent fractions were diluted to the final concentrations of 312.5-10000 mg/l each but in table1 are concentrations 3.12-100 grams per liter. There is 10 time error somewhere and this also appears in text LINE181 LC50 values of 7.81 g/l and in table 2 is 781 mg/l which is only 0.781 g/l. Moreover there are more numbers in text which not correspond to tables. Information about replications in bioassay and about individuals counts is missing in material and methods

Conclusions are vague formulated, practically missing.

In whole article I found some typos and merged words which need correction

Author Response

  1. In presented manuscript entitled "Linking volatile organic metabolites as produced by Purpureocillium lilacinum 6029 for the sustainable management of root-knot nematodes" authors analyzed nematocidal effect of various extracts obtained from P. lilacinum. Authors used non-typical substrate for cultivation Karanja deoiled cake, which should improve nematocidal activity as they reported previously. As this had probably high impact or achieved results I recommend adding this info to the title. On the other hand I am not sure that metabolites in extract were volatile, may be some of them, but not all so I suppose to remove this word from title.

 Response: The word “volatile” from the title has now been removed now as suggested. Thanks. Similarly, additional info has now been added in the title. Thanks for the kind suggestion.

  1. In introduction as well as in discussion authors deal a lot with various substrates and their effect on fungal metabolites. This is very interesting but this is not analyzed in article. Comparison of metabolites from traditional cultivation and karanja based method should be performed. Otherwise author must to focus on rather on metabolites themself than on cultivation substrate. Comparison of substrates is a weakness of manuscript. Whole work is more exploratotory than hypothesis driven, but it looks to be actual.

Response: The current study was an outcome of our work wherein we established the Karanja deoiled cake as novel substrate for the growth and pathogenicity of P. lilacinum (Sharma, A.; Sharma, S.; Mittal, A.; Naik, S.N. Statistical optimization of growth media for Paecilomyceslilacinus 6029 using non-edible oil cakes. Ann. Microbiol. 2014, 64, 515–520). Therefore, we highlighted this information in the introduction. But, we do acknowledge the concern of the reviewer about the irrelevance of this information in the discussion section and hence removed the said statements (line 241-247 in the original article). 

  1. Introduction is short and do not provide insight to metabolome of entomopathogenic/ nematopathogenic fungi.

Response: Although the chemical ecology of nematophagous fungi is still far from understood, we have tried to incorporate some examples of nematocidal metabolites being secreted by different groups of fungi (line 46-52 in the revised manuscript).

  1. Why authors incorporate morphological study and SEM? I consider it to be redundant. Shape of spores is not important when we are talking about metabolites. According my opinion it should be removed from article (from methods, results and discussion)

 Response: Thanks for your suggestion. We have removed the said study from the entire manuscript.

  1. In material and methods it is stated than solvent fractions were diluted to the final concentrations of 312.5-10000 mg/l each but in table1 are concentrations 3.12-100 grams per liter. There is 10 time error somewhere and this also appears in text LINE181 LC50 values of 7.81 g/l and in table 2 is 781 mg/l which is only 0.781 g/l. Moreover there are more numbers in text which not correspond to tables. Information about replications in bioassay and about individuals counts is missing in material and methods.

 Response: The authors acknowledge the keen observation of the reviewer and admit the discrepancy in mentioning the units. Concentrations are now uniformly defined in “mg/l” in the result section as well as in Table 1. Thanks once again.

  1. Conclusions are vague formulated, practically missing.

Response: Conclusions have been rephrased. Thanks.

  1. In whole article I found some typos and merged words which need correction

Response: The whole manuscript has been re-checked carefully and all typo errors and other mistakes have been rectified. Thanks.

Reviewer 3 Report

Dear Authors,

L23 - Sifnicant - Typo?

L27 - metabolitesproduced- Space should be there

L31 - giving 94.6% - space required

L34 - is it 3,5-Di-t-butylphenol?

L42 - ofhost - space required

L42 - preferences[1,2] - space between references and prefences.

L43 - accepted as an

L44 - This encompasses

**** Please check and add spaces. There are many similar mistakes through out manuscript. In addition, there should be a space between references ad last word of the sentence.

L157 - Figure number sould be added after "SEM images of P. lilacinum6029 showed spores present in optimized media"

Figure 1 came first. But you have put table 1 first. Please rearrange figures and tables according to the story you are telling.

Have you tried Ethy acetate alone as a control to meausre mortality ? And same thing has been done for other chemicals?

L165 - 5.0/l ethyl acetate showed maximal nematocidal activity? Is it true? Is it 50?

L167 - higest juvenile mortality 89.2% in 10g/L? i cannot see such data

L181 - Lc50 - 7.81. In your table 781.48. Please correct them carefully. All table 2 values are incorrect.

Funding and Conflict of interest should be revised appropriately.

Author Response

L23 - Sifnicant - Typo?

Response : Corrected

L27 – metabolites produced- Space should be there

Response : space added between two words

L31 - giving 94.6% - space required

Response : space added between two words

L34 - is it 3,5-Di-t-butylphenol?

Response : Yes. Corrected

L42 - ofhost - space required

Response : space added between two words

L42 - preferences[1,2] - space between references and prefences.

Response : space added between references and preference.

L43 - accepted as an

Response : Corrected

L44 - This encompasses

Response : Corrected

**** Please check and add spaces. There are many similar mistakes through out manuscript. In addition, there should be a space between references ad last word of the sentence.

Response : Spacing issue may be due to change in the MS-word version. However, whole manuscript is now corrected for proper spacing

L157 - Figure number sould be added after "SEM images of P. lilacinum 6029 showed spores present in optimized media"

Response : Reviewer 2 has asked to remove this part in his comments No 4. So we have removed this part from Materials and Methods and Results Discussion  

Figure 1 came first. But you have put table 1 first. Please rearrange figures and tables according to the story you are telling.

Response : Arranged properly. Figures one has been removed (as per the comments No 4 of Reviewer 2) so Table 1 will come first

Have you tried Ethy acetate alone as a control to meausre mortality ? And same thing has been done for other chemicals?

Response : Yes it was done with ethyl acetate and hexane as blank control and mentioned in Materials and Methods part Line 123-124 and Results Line 172-174 and in Table 1.

L165 - 5.0/l ethyl acetate showed maximal nematocidal activity? Is it true? Is it 50?

Response : It was a typo now corrected as 5,000 mg/L. Line 161

L167 - higest juvenile mortality 89.2% in 10g/L? i cannot see such data

Response : This data is already mentioned in Table under aqueous extract column Row No 5 and highlighted in red font.

L181 - Lc50 - 7.81. In your table 781.48. Please correct them carefully. All table 2 values are incorrect.

Response : All the values of Table 2 are now corrected

Funding and Conflict of interest should be revised appropriately.

Response : Funding and Conflict of interest have been revised appropriately.

Round 2

Reviewer 3 Report

Dear Authors,

Manuscript has been improved.